# Morphodynamic limits to environmental signal propagation across landscapes and into strata

Stephan C. Toby[1], Robert A. Duller [1✉], Silvio De Angelis[1] & Kyle M. Straub [2]

The sedimentary record contains unique information about landscape response to environmental forcing at timescales that far exceed landscape observations over human timescales. However, stochastic processes can overprint and shred evidence of environmental signals, such as sediment flux signals, and so inhibit their transfer to strata. Our community currently lacks a quantitative framework to differentiate between environmental signals and autogenic signals in field-scale analysis of strata. Here we develop a framework and workflow to estimate autogenic thresholds for ancient sediment routing systems. Crucially these thresholds can be approximated using measurements that are readily attainable from field systems, circumventing the low temporal resolution offered by strata. This work demonstrates how short-term system dynamics can be accessed from ancient sediment routing systems to place morphodynamic limits on environmental signal propagation across ancient landscapes and into strata.

[1] Department of Earth, Ocean and Ecological Sciences, University of Liverpool, 4 Brownlow Street, Liverpool L69 3GP, UK. [2] Department of Earth and Environmental Sciences, Tulane University, 6823 St. Charles Avenue, 202 Blessey Hall, New Orleans 70118-5698 LA, USA. ✉email: rduller@liv.ac.uk

A n environmental signal is a change in any physio-bio-chemical attribute of the Earth's surface and/or sediments in response to environmental forcing (tectonic, climatic, sea level, anthropogenic)[1]. Environmental signals may be found in time series of, for example, isotope data, palynological, palaeoecological, petrophysical or petrographic data, or indeed any measurable, observable time-dependent property linked to environmental forcing. A key measurable attribute that links environmental forcing, Earth surface processes and strata is sediment flux, and we focus our efforts on this attribute. The sedimentary record offers important insights into Earth System response over the full duration of past episodes of environmental change over time periods of Earth's history that far exceeds records from modern scientific instrumentation[2]. However, the physical nature of the sedimentary record is not an exclusive product of environmental forcing but is also a product of internal system processes and self-organized dynamics (i.e. autogenic processes) of sediment transport systems[3–5]. As environmental forcings operate at similar timescales to autogenic processes, environmental signals can become obscured or shredded[4], preventing their transfer to the stratigraphic record[4,6–9]. A general framework that is frequently applied to ancient field-scale sediment routing systems (SRSs) is landscape or fluvial diffusion[1,10], and the estimation of associated basin response timescales, $T_{eq}$[11]. This response timescale is then compared to the duration or periodicity of a particular environmental forcing to assess whether the environmental signal is likely to be buffered or not[10,12,13]. This diffusive approach, although insightful and eminently applicable to field-scale systems, implicitly assumes a degree of spatio-temporal averaging of stochastic autogenic processes[1]. This assumption leads to a loss of predictive capability when evaluating spatio-temporal limits of both singular and periodic environmental signal propagation through SRSs, and to strata. However, stochastic autogenic processes are inherent to three-dimensional sediment transport systems and set a lower threshold for the propagation of sediment flux signals through SRSs and to the stratigraphic record[1,3,4,8,14–18]. Thus, any theoretical framework must incorporate autogenic processes, but also be flexible enough so that it can be applied directly to field-scale systems. This lower threshold or autogenic threshold function (ATF) was defined by Toby et al.[7] as the maximum rate of autogenic flux (or volume change) that a sediment transport system can experience over a specified time window. Toby et al.[7] demonstrated that the maximum rate of autogenic flux decays as an exponential function of the timescale of measurement (Fig. 1). Any combination of sediment flux signal magnitude and duration that plots above the ATF should generate a detectable signal in the preserved strata, while those that plot below the ATF will be absent or intermingled with autogenic signals of similar magnitude in the preserved strata (Fig. 1). The potential applicability of this model to ancient field-scale SRSs remains limited given the need for high temporal and spatial resolution volumetric data, akin to the laboratory experiments from which the ATF was derived. Here we first set out a workflow to approximate the ATF for ancient field-scale systems that circumvents the need for the exquisite time resolution of strata. Secondly, we demonstrate the applicability of this approach using two ancient field-scale systems: a single-segment SRS in Greece[19]; and a multiple segment SRS in the Spanish Pyrenees[20]. The approach outlined here offers a crucial null hypothesis to field scientists wishing to explore strata for signatures of Earth's past response to environmental forcing.

## Results

**Theoretical development.** To generate an ATF for field-scale systems a timescale of autogenic saturation and a magnitude of autogenic sediment fluxes must be incorporated[1,4,7]. Sedimentation rates generally decrease with increasing time window of measurement[21] as stochastic autogenic fluctuations in these rates saturate at a rate equal to the long-term aggradation or subsidence rate $(r)$[1,7,22]. The timescale at which this occurs can be estimated from channelized strata as a compensation timescale $(T_c)$: $T_c = H_{max}/r$, where $H_{max}$ is the maximum topographic roughness; maximum channel depth in this case[22]. In a physical sense, $T_c$ can be thought of as the maximum time necessary to bury a particle to a critical depth so that it is no longer susceptible to erosion, and represents an estimate of the maximum timescale of autogenic organization in stratigraphy[1,22]. This concept does not refer exclusively to channelized terrestrial environments. The compensation timescale can be cast more generally as $T_c = l_{max}/r$, where $l_{max}$ represents the maximum topographic roughness length scale of any chosen feature at the Earth's surface[22]. Therefore, $T_c$ can be determined for any environment where topographic features with roughness $l_{max}$ migrate over a sediment surface that is net depositional over the long term (see Supplementary Note 1). However, we note that the detrimental effects of autogenic processes, for the preservation of environmental signals in strata, will be most pronounced in channelized (e.g. fluvial and deepwater channel-fan) environments.

To enable direct comparisons between laboratory- and field-scale systems, volumetric sediment fluxes $(Q)$ are made dimensionless by dividing by the long-term mean sediment supply rate $(Q_{in}$, e.g. $Q^* = Q/Q_{in})$. Timescales are also made dimensionless by dividing time $(t)$ by $T_c$ $(T^*$, using an asterisk consistently for dimensionless variables). We use the exponential form of the ATF[7] (see Supplementary Note 2), $Q_a^* = Q_0^* e^{-bT^*}$, where $Q_a^*$ is the dimensionless autogenic flux and $Q_0^*$ is the dimensionless maximum autogenic flux at $T^* = 0$, to generate an ATF approximation that can be applied to ancient field-scale SRSs (Fig. 1). We set out to constrain the values of $Q_0^*$ and $b$ that define the exponential ATF function (Supplementary Note 3 and 4). Two or more combinations of $Q_a^*$ and $T^*$ are sufficient to constrain the value of $Q_0^*$ and $b$ and the challenge here lies in finding these combinations given the limited spatial and temporal resolution of ancient field-scale SRSs.

We begin by splitting the long-term mean sediment input flux, $Q_{in}^*$ into two components: a maintenance flux $(Q_{acc}^*)$ and the long-term (i.e. $T^* > 1$) bypass flux $(Q_{bp(L)}^*)$ (Fig. 2). The maintenance flux is a depositional flux required to fill accommodation produced by subsidence and/or relative sea-level rise. For simplicity, we assume that the rate of accommodation generation, and so $Q_{acc}^*$, is constant through time enabling $Q_{acc}^*$ to be calculated from the plan-view area $(A)$ of a segment of a SRS and long-term aggradation rate: $Q_{acc} = A \cdot r$. To approximate the value of $Q_0^*$ we assume that at very short timescales $(T^* \to 0)$ an environment has the capacity to retain all of the sediment delivered to it[23], i.e. $Q_0^* = Q_{bp(L)}^* = Q_{in}^* - Q_{acc}^*$. The rate of decay of $Q_a^*$, the exponential ATF function, is then described by the decay constant $b$ and asymptotes on approach to zero. We approximate the timescale at which $Q_a^*$ approaches zero as the time necessary for $Q_a^*$ to be reduced to 5% of its maximum value $(T_{95})$ (Fig. 1). Rearranging the exponential ATF function we find an expression for the autogenic decay constant $b$ that utilizes the timescale for the 95% reduction in $Q_a^*$ $[b = -\ln(0.05)/T_{95}^*$, where $T_{95}^* = T_{95}/T_c]$. Theory suggests that field-scale autogenic fluxes often saturate at $T_c$, specifically in systems where channels avulse at timescales significantly less than $T_c$[24]. Assuming $T_{95}$ is well approximated by $T_c$ $(T_{95}^* = 1)$ this gives a value of $b \approx 3$ (Supplementary Note 3 and Table S1). While the ATF was constructed with channelized environments in mind, we note that all depositional environments contain an element of autogenic flux, which gives rise to temporally incomplete strata at the finest

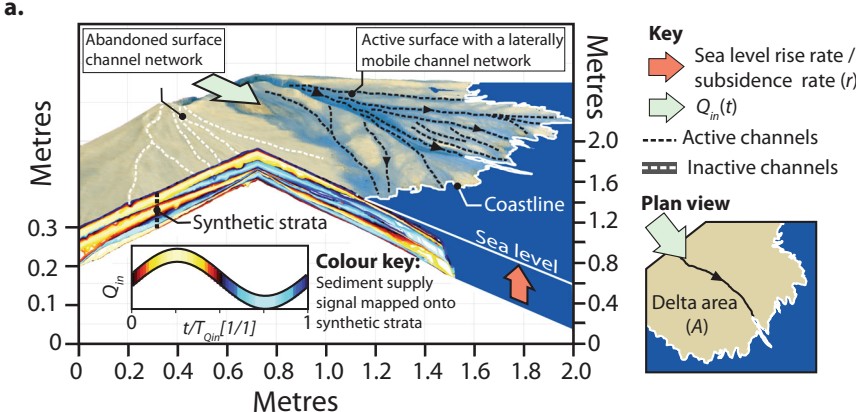

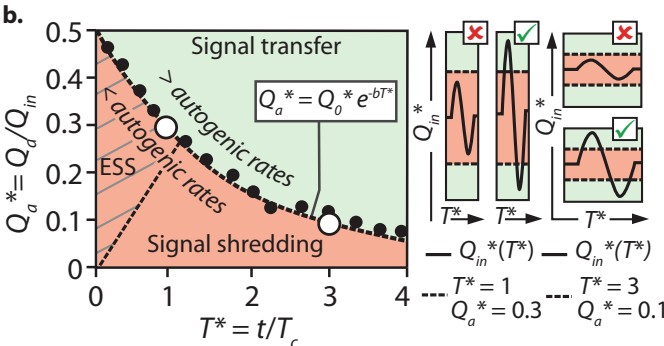

**Fig. 1 The natural autogenic behaviour of an experimental delta system. a** Topographic scan and co-registered digital image of experiment TDB-20 at run time 532 h dissected to show underlying synthetic strata. Warm and cool colours represent strata deposited during a peak and trough of sediment supply respectively. **b** Diagram outlining the autogenic threshold function (ATF); the position of maximum autogenic rates as a function of time (dashed black line). The ATF separates regions where sediment input signals, $Q_{in}*(T*)$, would be shredded by autogenic processes (salmon red) and a region where $Q_{in}*(T*)$ would be transferred to strata (green). White circles on the ATF diagram represent two example threshold conditions as depicted on the right. Each example shows how two different $Q_{in}*(T*)$ conditions are either shredded (signal remains in red) or transferred to strata (signal surpasses threshold to occupy green). Note that in this way the ATF can assess the likelihood of any combination of magnitude–period $Q_{in}*(T*)$ being transferred to strata. Region of grey diagonal line fill represents phase space where Earth surface signal (ESS) is detectable in the active layer (i.e. geomorphological change) but not in strata. Diagram generated using the data set of Toby et al.

timescales of discretization. Therefore, we anticipate that an ATF can be constructed for other environments at different temporal and physical scales[7,17].

**Experimental validation**. To test the validity of the above workflow we estimate the values of $Q_0*$ and $b$ from physical experiments[25,26] using the field-ATF approximation and compare these values to those measured from the high-resolution datasets of the same experiments (Supplementary Note 3–5). We note that the experimental data suggest that $Q_a*$ reduces to $0.05Q_0*$ at a timescale 6–8$T_c$, which would suggest $b \approx 0.4$–0.5. This extended timescale is due to the long avulsion timescales ($T_c$) of the experimental delta channels, specifically late in the control experiment, some of which were similar in duration to $T_c$. When constructing the ATF from data earlier in the experiment, which was unaffected by excessively long avulsion timescales and had similar water depths to our experiments with temporally variable sediment flux, the value of $Q_a*$ reduces to $0.05Q_0*$ at a timescale $T_c$ supporting our assertion that autogenic fluxes saturate at $T_c$ (see Supplementary Note 5). Furthermore, in field-scale systems[27], avulsion timescales are generally $T_A \approx 0.006T_c - 0.06T_c$ (Supplementary Note 5 and Table S2). As such we expect that autogenic fluxes in the vast majority of field scale

systems will saturate at timescales closer to the theoretical minimum of $T_c$, which supports the use of a constant value of $b \approx 3$ for the application of the ATF framework to field-scale systems. We find that both experimental ATF and field-approximated-ATF from the same experiments generate similar values for $Q_0*$ (Table 1) and thus validates this component of our approach.

**Field application**. Following our workflow, we can approximate the ATF for ancient field systems with knowledge of $T_c$ and $Q_{in}$. Estimating a conservative value of $T_c$ involves the measurement of minimum values of $r$ and maximum values of $H_{max}$ from strata[8,28]. The difficulty will always remain in the estimation of $Q_{in}$[29,30] but this can be overcome to some extent through the use of empirical equations of sediment flux predictors such as the BQART method[31,32], or with knowledge of the long-term volumetric sediment mass balance of closed SRSs or SRS segments. We apply our field ATF to two field systems: the Pleistocene Kerinitis Delta System (KDS) in Greece and the Eocene Escanilla Sediment Routing System (ESRS) in northern Spain.

The KDS represents a Gilbert-type fan delta that accumulated in a normal fault-controlled depocenter (hanging-wall) in Lake Corinth during the Pleistocene[19]. Here we focus on the fluvial topsets of the

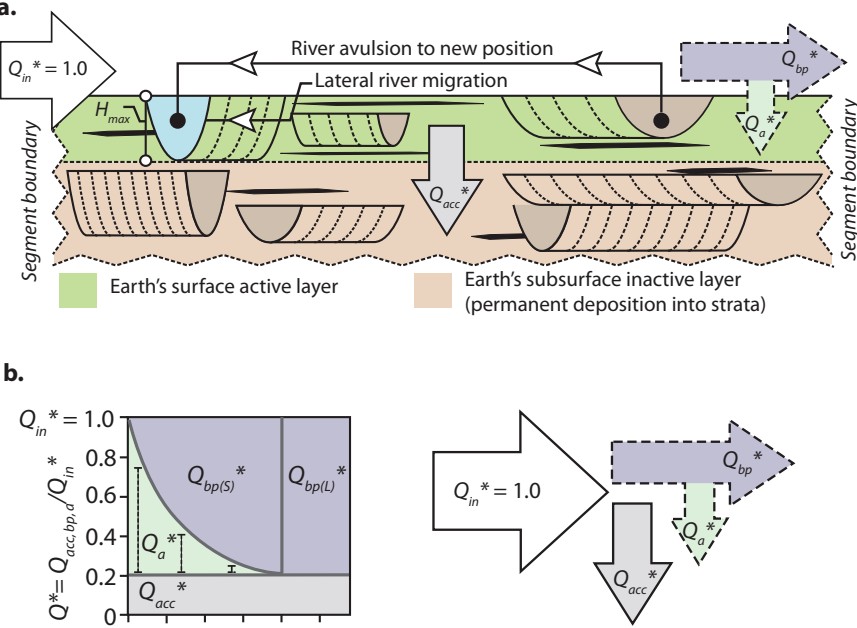

**Fig. 2 The sediment flux budget of a sediment routing system. a** Conceptual diagram showing how input sediment flux ($Q_{in}^*$) is partitioned within a channelized sediment routing system. **b** Graphical representation of how $Q_{in}^*$ is partitioned into maintenance flux ($Q_{acc}^*$), autogenic flux ($Q_a^*$) and bypass flux ($Q_{bp}^*$) for increasing time windows, $t$. The additional bracketed subscripts to $Q_{bp}^*$ represent short-term (S) and long-term (L). The maximum value of $Q_a^*$ decreases as a function of the time window and asymptotes towards zero at $T^* \geq 1$ or $\geq T_c$, (compensation timescale). Note that this timescale also defines the boundary between $Q_{bp(S)}^*$ and $Q_{bp(L)}^*$.

**Table 1 Experimental parameter values for the ATF and field-approximated parameter values of the ATF for the same experiments.**

|  | TDB-12 stage 2 | TDB-13 stage 2 | TDB-13 stage 1 |
|---|---|---|---|
| Experimental ATF | $Q_a^* = 0.33e^{-0.49T^*}$ | $Q_a^* = 0.46e^{-0.38T^*}$ | $Q_a^* = 0.35e^{-0.18T^*}$ |
| Experimental $Q_0^*$ | 0.33 | 0.46 | 0.35 |
| Field-approximated $Q_0^*$ | 0.47 | 0.34 | 0.27 |
| Experimental $b$ | 0.49 | 0.38 | 0.18 |
| Field-approximated $b$ | ≈3 | ≈3 | ≈3 |

See text for discussion on the value of $b$ for experimental and field-approximated ATF.

KDS when constructing the KDS-ATF (Fig. 3a), while the foresets represent the bypassed sediment volume. The mean sediment flux ($Q_{in}$) to the delta is unknown and so here we use a conservative range of values defined by the fractional retention of $Q_{in}$ in the subsiding basin (e.g. $Q_{acc}/Q_{in} = 0.1, 0.5, 0.9$) and a field-calculated $T_c$ of ≈5.7 ky to define the position of the KDS-ATF (Fig. 3b). SRSs in the Gulf of Corinth were particularly susceptible to climate cycles in the Pleistocene[33], with speculation that the KDS contains evidence of obliquity-scale (40 ky) environmental forcing[19]. Inspired by BQART-estimates for obliquity-scale sediment flux signals in the Gulf of Corinth and elsewhere[32,34], we test whether a 30% change in input sediment flux ($0.3Q_{in}$) would exceed the calculated KDS-ATF. We find that obliquity- and precession-scale sediment flux signals are likely to have exceeded the KDS-ATF (Fig. 3b), supporting the assertion that the KDS succession preserves signatures of these environmental forcing[19].

The ESRS can be traced down-system over 200 km and is divided into five segments (Figs. 4a and 5). Michael et al.[20] estimated the total flux into each segment ($Q_{in}$) and the flux retained within each segment ($Q_{acc}$) for a time interval between 39.1-36.5 Ma (Table 2). This data enables the long-term bypass

flux ($Q_{bp(L)}^*$) to be calculated at the down-system boundary of each segment (Supplementary Note 6), which is used as $Q_{in}$ for the next segment in the SRS (Fig. 4b). Using this information, in combination with estimates of $T_c$ from published values or $r$ and $H_c$ (Table 2) we determine the ATF of each segment within the ESRS (Fig. 6). We use the field ATFs to evaluate the potential of each segment to store orbital-scale environmental forcing signals with durations of 20, 40 and 100 ky, again assuming a magnitude of change $0.3Q_{in}$ for each. Calculated positions of the ATF for each segment predict that environmental signals of 20 ky cyclicity are unlikely to be recorded in the strata of any of the segments, whereas environmental signals of 40 and 100 ky cyclicity are likely to be preserved in strata of the Sis and Gurb segments (Fig. 6). The 40 ky cycles are not estimated to be preserved in the Graus and Aínsa segments because $T_c$ is much longer in these fluvial systems compared to the alluvial fans of Sis and Gurb segments (Fig. 6 and Table 2). The value of $T_c$ in the Escanilla–Graus segment is particularly long, which may prevent the stratigraphic transfer of the 100 ky cycles in this part of the ESRS. We note that our framework does not predict an ATF for the Jaca segment given that there is no bypass flux for the most distal segment.

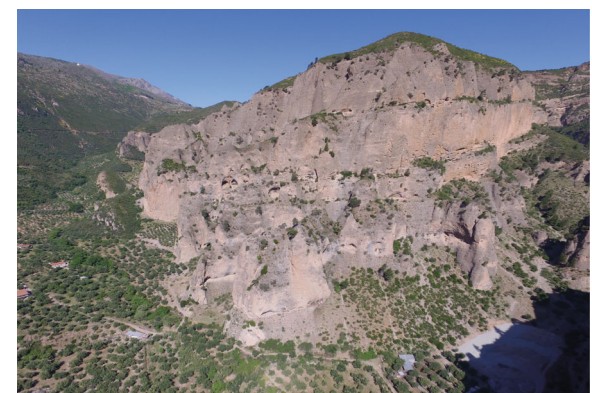

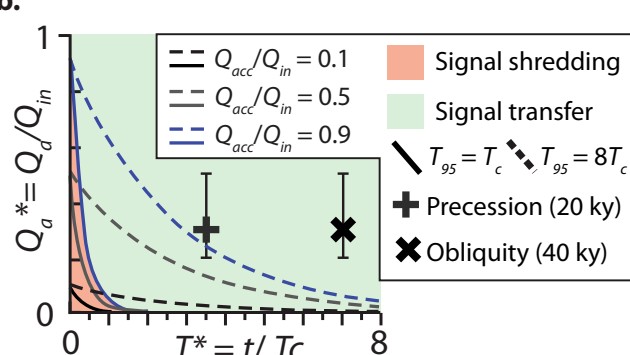

**Fig. 3 The Pleistocene Kerinitis Delta System (KDS). a** Oblique aerial image of the topsets of the KDS developed in the hanging wall of the Pirgaki–Mamoussia fault (planar-inclined surface to the left). The height of the outcrop is ~500 m. Image courtesy of Dr Bonita Barrett (Equinor). **b** ATF diagram for the KDS showing two estimations for the position of the autogenic threshold function. A scenario where $Q_a^*$ approaches zero at $T_{95} = T_c$ (solid lines) and a scenario where $T_{95} = 8T_c$ (dashed lines). Allogenic sediment flux signals have an assumed magnitude of $0.3Q_{in}$, and a range between 20 and 50% of $Q_{in}$ based on estimates of Milankovitch-scale sediment supply variations (Blum and Hattier-Womack[34]; Watkins et al.[32]).

## Discussion

**Key uncertainties in the field-approximated ATF.** Our formulation of the time-dependent field-ATF builds on established work that predicts an upper temporal limit to the ability of autogenic processes to obscure allogenic signals. Application of the field-ATF relies upon estimates of $Q_{in}$, $Q_{acc}$ and $T_{95}$. Reasonable values of $Q_{in}$ can be obtained from empirical equations or a basin-wide sediment volume-balance[29,30]. Given the spatial incompleteness of geological data and the open character of many basins, underestimating $Q_{in}$ is likely, which would lead to an underestimation of the ATF and an overestimation of the system's storage potential. Estimates of $Q_{acc}$ depend on the resolution of age constraints in strata and the ability to reconstruct a plan-view area of the SRS. The largest uncertainty in the field-ATF framework may reside in the estimation of $T_{95}$ and therefore the exponential decay coefficient $b$. We expect a lower limit of $T_{95}$ to be set by $T_c$, but $T_{95}$ may be longer in scenarios where autogenic surface processes (i.e. avulsions) occur over timescales of the same order as $T_c$. For example, given the short $T_c$ in the KDS, it is possible that $T_{95} > T_c$, which would generate lower values of $b$, whereas the much longer values of $T_c$ in the ESRS would generate a larger value of $b$. Regardless, a challenge remains to define the temporal limit of autogenic flux saturation (i.e. long-term sedimentation rates persist), which will reduce uncertainty in the position of the field-ATF.

**Temporal limits of signal propagation through SRSs.** There are two key timescales that emerge in the study of SRSs: the equilibrium timescale ($T_{eq} = L^2/k$ where $L$ is system length, and $k$ = transport or diffusion coefficient) and the compensation timescale ($T_c = H_{max}/r$). Whereas $T_{eq}$ describes the timescale for regrading of surface topography, $T_c$ describes the timescale for the construction of basin-wide strata with a thickness equal to the largest roughness element of a system (i.e. maximum channel depth, $H_{max}$). Given that both timescales emerge as a consequence of the long-term spatial distribution of sediment deposition, their absolute values should be of the same order of magnitude for a given system[1]. Therefore, when each of these timescales is surpassed, basin-wide topography (i.e. $T > T_{eq}$) and strata (i.e. $T > T_c$) are set by allogenic forcing. Given the large uncertainty associated with the value of the diffusivity coefficient ($k$) for ancient SRSs[35],

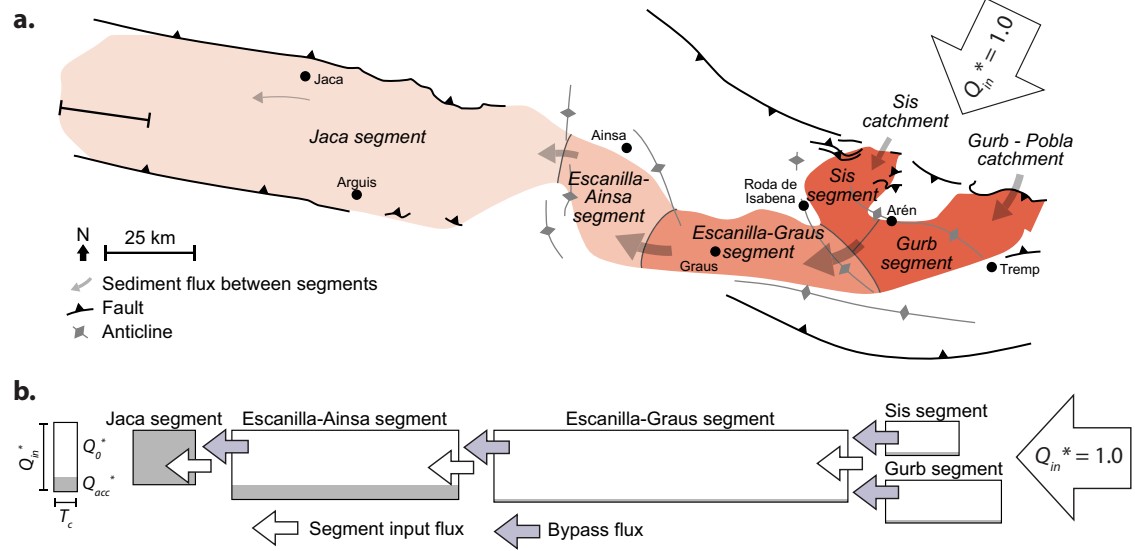

**Fig. 4 The Escanilla Sediment Routing System (ESRS) between 39.1 and 36.5 Ma. a** Major structural features and segments of the ESRS. Figure modified from Michael et al.[20]; **b** Schematic diagram showing $T_c$ (width of the box), $Q_{in}^*$ (height), $Q_{acc}^*$ (black) and $Q_o^*$ (white) for each segment of the ESRS. See Table 2 for parameter values. Note that the segment input flux is equal to the bypass flux of the preceding segment.

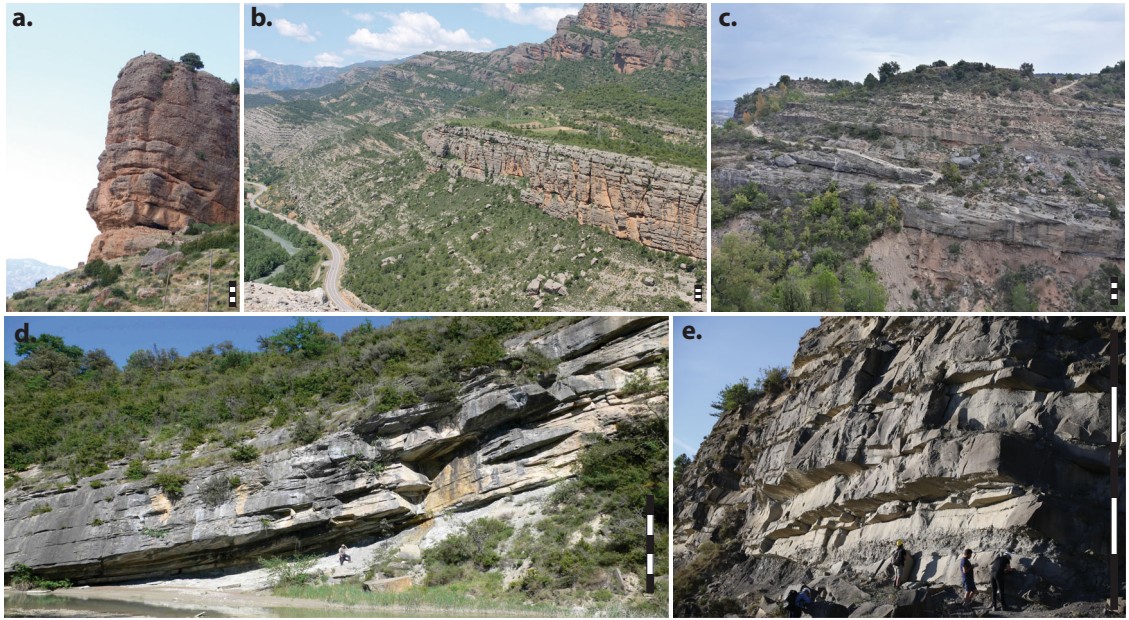

**Fig. 5 Outcrop photographs of strata from each segment of the Escanilla Sediment Routing System.** Monstor alluvial fan succession, Gurb segment (**a**, **b**); Escanilla fluvial succession, Escanilla–Graus segment (**c**); Guaso 1 turbidite fan succession, Escanilla–Ainsa segment **d**; Ainsa turbidite fan succession (**e**). The scale bar at the bottom right of each image is 10 m. Images courtesy of Professor Kevin Pickering (University College London), Dr Miquel Poyatos-Moŕe (Universitat Autònoma de Barcelona) and Dr Rhodri Jerrett (University of Manchester).

**Table 2 Key data used for the calculations of transfer thresholds of the Kerinitis Delta System[19] and the Escanilla sediment routing system[20].**

|  | Kerinitis Delta | Escanilla Sediment Routing System | | | | |
|---|---|---|---|---|---|---|
| SRS segment | — | Gurb | Sis | E-G | E-A | J |
| Member | — | Montsor units 1 & 2 | Sis units 1 & 2 | Mid. Escanilla | Mid. Escanilla | Upper Hecho |
| Environment | GFD | AF | AF | F | F | SM, DML |
| $Q_{in}$ (km³/My) | $Q_{acc}/0.1 \leq 0.9$ | 106 | 132 | 222 | 214 | 168 |
| $Q_{acc}$ (km³/My) | 0.008 | 7 | 8 | 9 | 46 | 168 |
| $Q_{acc}$* | $1 \times 10^{-5}$–$3 \times 10^{-4}$ | 0.06 | 0.06 | 0.04 | 0.21 | 1 |
| Thickness (m) | 800 | 140 | 220 | 160 | 250 | 1000 |
| Length (km) | 22 | 35 | 25 | 60 | 30 | ≥125 |
| $r$ (m/ky) | 1.77 | 0.054 | 0.085 | 0.062 | 0.096 | 0.38 |
| $H_c$ (m) | 10 | 5 | 5 | 17.5 | 17.5 | 40 |
| $T_c$ (ky) | 5.7 | 93 | 59 | 284 | 182 | 104 |

SRS segment abbreviations: *E-G* Escanilla–Graus segment, *E-A* Escanilla–Aínsa segment, *J* Jaca segment. Environment abbreviations: *GFD* Gilbert fan delta, *AF* alluvial fans, *F* fluvial, *SM* shallow marine, *DML* deep marine lobes. Values of $H_c$ are from Vincent[60], Labourdette[61] and Bayliss and Pickering[62]. KDS dimensions and subsidence rates are from Barrett et al.[19] and channel depths are from Backert et al.[63].

estimates of $T_{eq}$ are generally within an order of magnitude. The uncertainty in the estimation of $T_c$ is, however, much less severe as it can be directly estimated from field data (i.e. channel depth and subsidence rate) and the uncertainty in estimations of $Q_{in}$ is also less than current (diffusive) frameworks.

Timescales $T_{eq}$ and $T_c$ represent singular temporal limits for signal propagation and preservation[8,12,22] and so both will only provide information about the timescale of response relative to the event timescale or periodicity. A key difference lies in the fact that the formulation of $T_{eq}$ implicitly averages lateral stochastic system dynamics[11] while the formulation of $T_c$ explicitly incorporates stochastic system dynamics[22,36]. This has enabled us to extend the singular stochastic temporal marker to a continuous stochastic threshold at timescales $T < T_c$, incorporating the temporal dependence of autogenic magnitude[7]. The key implication of the field-ATF is that it provides field scientists with a tool to predict the presence (or absence) of any magnitude–period combination of input sediment flux signal in strata.

The formulation of $T_{eq}$ for sedimentary basins[11] incorporates surface elevation change only and therefore only involves an active layer. Therefore, all signals of environmental forcing that induce a surface topographic change will be directly transferred to permanent strata, unless it undergoes erosion induced by a subsequent environmental forcing episode that is greater in magnitude than the previous environmental forcing. Autogenic processes have no role in signal propagation and storage in a fluvial diffusion framework, and there is no distinction between active layer and strata. Therefore, an environmental sediment flux signal that is buffered ($T < T_{eq}$)[11,13,10] in the stratigraphic sense must, by definition, also be buffered in the Earth surface sense, and so signal propagation is halted. The autogenic environmental signal shredding framework of Jerolmack and Paola (2010)[4] also deals exclusively with the active layer when defining an autogenic saturation timescale ($T_x \approx L^2/q_{in}$, where $L$ is system length and $q_{in}$ is sediment input rate) or temporal threshold for surface signal propagation. To exemplify the disconnect of both $T_{eq}$ and $T_x$

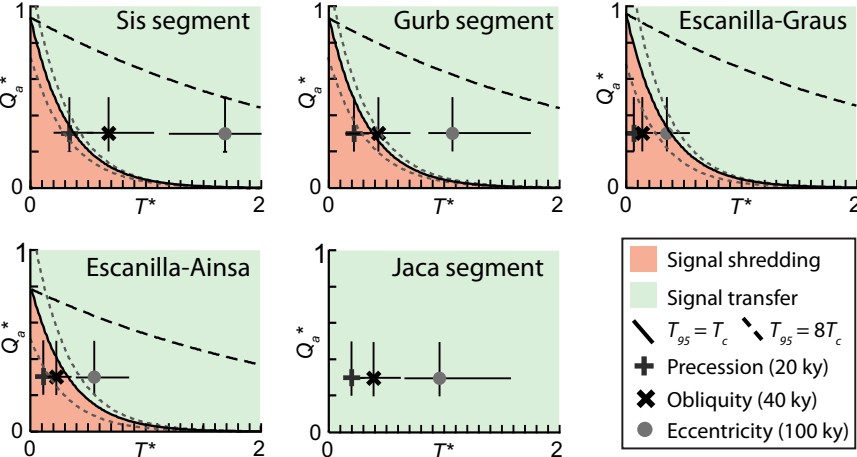

**Fig. 6 Estimates of the autogenic threshold function (ATF) for each segment of the ESRS segments based on data from Table 2.** The position of Milankovitch-scale signals ($t = 20$ ky, $t = 40$ ky and $t = 100$ ky), assuming a change in sediment input flux of ~$0.3Q_{in}$ (range $0.2Q_{in}$–$0.5Q_{in}$) are plotted on each ATF to establish whether these signals are likely to be shredded at the Earth surface active layer or transferred to strata. Dashed grey lines represent a standard upper and lower envelope of the ATF curve based on the parameter ranges presented in Table 2.

from strata, both can be calculated in a sedimentary basin that is undergoing zero subsidence. The ATF approach as outlined here definitively links to strata through the definition of $T_c$, whereas $T_{eq}$ and $T_x$ represent regarding timescales of the Earth's surface. As a result, the field-ATF approach bridges the temporal gap between the shorter-term sediment fluxes of the Earth's active layer and long-term fluxes associated with basin-filling timescales and the generation of permanent strata.

**The critical role of the Earth surface active layer.** The incorporation of stochastic dynamics into the field-ATF implicitly requires that a fraction of $Q_{in}$ contributes to an autogenic sediment flux, $Q_a$. This autogenic flux operates within the temporal bounds of $T_c$ and the spatial bounds of the largest roughness length scale in the system (e.g. $H_{max}$). Therefore, $Q_a$ defines a dynamic Earth surface active layer[37,38] that contains information pertaining to the complexity and nature of strata within the autogenic timescale range[39,40]. Importantly, the active layer can act as a transient store of environmental information, and as both inhibitor and conveyor of environmental signal propagation[4]. Practically, what this means is that evidence of a particular sediment flux signal may not be stored in the stratigraphy of one SRS segment, yet the signal may still have propagated across the Earth surface active layer to the next SRS segment–segment. If the next segment has a sufficiently low ATF, that same signal may exceed the threshold of that segment and be stored in its strata. This could be the case for an eccentricity-scale sediment flux signal in the ESRS where it is predicted to be stratigraphically shredded in the Escanilla–Graus segment while strata in the Escanilla–Aínsa segment (Fig. 6) would have the ability to preserve such a signal. Our observation that environmental sediment flux signals can be stratigraphically shredded in one segment yet stored in the next highlights the need to understand how the ATF might change in a single segment of a single basin. Given that a higher local rate of tectonic subsidence is associated with a lower value of $T_c$ (and vice versa), subsidence is a first-order control on whether or not an environmental signal will be stored in strata. Dimensionally, this is because: (1) the timescale at which the ATF approaches zero decreases with increasing subsidence rate (assuming constant $H_{max}$), and (2) the threshold magnitude decreases with increasing subsidence rate, as a larger proportion of $Q_{in}$ goes into $Q_{acc}$ relative to into $Q_a$, and so the likelihood of signal preservation increases. This behaviour is demonstrated to

some extent by the application to the ESRS (Fig. 3), where $Q_{acc}$ and $T_c$ are calculated for each segment, fixing the position of the ATF.

Our ATF approach does not incorporate the possible transformation of an environmental sediment flux signal as it propagates through the active layer of a SRS. In other words, we assume that the duration of an environmental signal when entering segment 1 will have the same unaltered duration when entering segment 2, irrespective of distance travelled. Depending on the length scales involved and the duration or period of the environmental forcing, sediment–storage–bypass–release in and above the active layer can have the effect of increasing the duration of the environmental signal and reducing its magnitude further as it propagates through a SRS[12]. We note that, although this effect is a deterministic outcome of diffusion, the degree to which this takes place in field-scale systems that involve stochastic autogenic flux is unknown, as is whether these effects are sufficient to significantly alter the position of a signal on the ATF. The ATF framework predicts that apart from signals that exceed the ATF, high-frequency ($T^* \ll 1$) sediment flux signals with a particularly high rate of flux change (region ESS; Fig. 1b) can induce an active layer or geomorphic response[7]. This happens because autogenic surface processes cannot redistribute sediment across the active layer at the required rate to maintain an equilibrium landscape with respect to the signal of sediment flux. Therefore, this class of signal, especially if $Q_a^* > 2.5$ (Fig. 1b), is still transmitted through the active layer of a SRS (see Supplementary 2). These represent "system clearing events" as envisaged by Jerolmack and Paola[12] for the geomorphic active layer, while events with $T^* < 1$ but exceed the ATF represent stratigraphic system clearing events. Glacial–interglacial cycles or rapid-onset global warming and global cooling events[32,41] are likely to trigger such high acceleration environmental signals and system-wide sediment dispersal, which will undoubtedly leave a mark in the landscape[42], but the stratigraphic transfer is not guaranteed unless they are of sufficient magnitude to surpass the ATF at these shorter timescales[43]. We anticipate that this class of signal is likely to undergo severe attenuation as they propagate through a SRS, and so the initial value of $Q_{in}$ must be large enough to ensure that the modified, attenuated signal plots above the ATF of the next segment, and become preserved in strata. Conversely, slowly changing environmental signals that do not exceed the ATF are overprinted by autogenic sediment flux signals in the active layer and are thus indistinguishable from autogenic flux variations upon exit of a SRS or SRS segment.

The active layer will always tend to smooth out (diffuse) or mix up (shred) environmental signals to a greater extent with increasing distance travelled through a SRS. The ability of strata to store environmental signals often does not decrease down-system as it depends on gradients in subsidence (or aggradation) down-system. We note that both $T_{eq}$ and $T_x$ offer an approximate scaling between input sediment flux and system length in one-dimensions, and so estimates of their value will always be within an order of magnitude. However, $T_{eq}$ remains the only tool with which to predict how environmental signals are dissipated and transformed as they propagate through the active layer of a SRS. A promising path forward is to expand the construction of $T_x$ to three-dimensional landscapes, which means understanding the fundamental mechanics of sediment trajectory through a SRS[44,45].

**Sediment flux signals in deep marine strata**. If signals can propagate through a SRS and are not shredded by autogenic processes in the active layer then environmental signals will be stored in deep marine strata. In deep marine strata of the ESRS Escanilla–Aínsa segment, precession-scale, obliquity-scale and eccentricity-scale environmental sediment flux signals are identified[46,47]. Within the confines of our ATF framework, the prediction that such signals should be stratigraphically shredded by upstream segments (Fig. 6) implies that the origin of these signals contained within strata are either autogenic in origin or represent environmental sediment flux signals whose magnitude remained sufficiently high and was therefore capable of propagating through the active layer of segments of the ESRS (region ESS; Fig. 1b). Similarly, the interaction between sediment flux and water flux could induce a nonlinear response that might amplify sediment flux signals, and enhance environmental sediment flux signal propagation across SRSs and to marine segments[9,48,49]. However appealing it may be to interpret deep marine signals on the basis of environmental signal propagation through terrestrial segments a fundamental question is how environmental signals, if they successfully reach the shoreline, cross this energy gate and traverse the shelf with minimal storage and shredding[6,45]. Furthermore, the storage and liberation of sediment on the shelf will be affected by sea-level oscillations, which may overprint any incoming environmental sediment flux signal[50,51]. A fundamental question still remains as to how different environmental signals propagate through the active layer of landscapes or SRS and if they can propagate to the shoreline and beyond[45].

**SRS sensitivity**. We re-emphasize that it is the characteristics of a specific landscape or SRS that determines its response to environmental forcing and its ability to transfer environmental signals to strata[52–54]. For example, where links between catchment and basin are short and direct and basin subsidence is relatively high, such as the normal-fault bounded KDS (Fig. 3), the likelihood of environmental signal transfer to strata is far greater. Given further consideration to the control of tectonic subsidence on environmental signal preservation, we expect different basin types to store environmental signals in different ways. For example, subsidence at passive margins often increases from source to sink and so the storage potential of an environmental signal will be less in proximal regions because of the limited accommodation production, whereas strata in proximal regions of foreland basins and hanging-wall basins would favour environmental signal storage. The value of $T_c$ also depends crucially on flow depth, (i.e. $T_c$ = flow depth/subsidence rate). Down-system changes in flow depth are fundamentally linked to how water discharge changes down-system (e.g. distributive or tributive channel networks[55])

and to backwater effects[56,57]. An interesting avenue of future work for field-ATF application will be to determine the value of $T_c$ as a function of down-system distance for different down-system channel network patterns and subsidence regimes.

The field-ATF approach as outlined here presents a volumetric assessment of environmental signal propagation and transfer in SRSs. A key area for further investigation is whether predictions of the field-ATF can be tied to distinctive patterns in sedimentary architecture or vertical organization of units at outcrop-scale[58,59]. In this respect, the field-ATF framework provides a much-needed null hypothesis that may guide future field investigations, alongside other proxies that may capture environmental sediment flux such as changes in paleo-slope and shoreline position. The flexibility of the field-ATF presented here means that it can be used to assess the stratigraphic transfer of any combination of period/duration and magnitude. Although we focussed on orbital timescales (i.e. >20 ky), the framework also provides an insight into SRS response, and potential signal transfer, related to punctuated high-magnitude environmental signals similar in magnitude to those of the present day[2,43].

## Data availability

The experimental data used in this study to validate the theoretical framework can be downloaded from the SEAD Data Repository: Li, Q. and Straub, K.M. (2017) TDB_12_1, SEAD, https://doi.org/10.5967/M03N21GX; Li, Q. and Straub, K.M. (2017) TDB_13_1, SEAD, https://doi.org/10.5967/M07D2S7Q.

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

## Acknowledgements

This study was supported by the National Science Foundation (grant EAR-1424312) and the Natural Environment Research Council (NERC) EAO Manchester-Liverpool Doctoral Training Partnership (grant NE/L002469/1). We thank the members of the Tulane Sediment Dynamics and Stratigraphy Lab for providing access to the experimental data. We thank two reviewers for helping us clarify the concepts presented in this manuscript.

## Author contributions

S.T.C., R.A.D., K.M.S. and S.D.A. compiled and analysed the data. R.A.D. and K.M.S. conceived of this study; S.T.C., R.A.D. and K.M.S. developed the theory, interpreted the results and wrote the manuscript.

## Competing interests

The authors declare no competing interests.
