## [Peer Review File · Nature Communications]

Morphodynamic limits to environmental signal propagation across landscapes and into strataReviewers' Comments:

Reviewer #1:

Remarks to the Author:

Dear Editors and Authors,

Toby and colleagues have recently proposed a novel approach to quantify whether or not certain magnitude-frequency combinations of sediment flux signals are likely to be preserved in stratigraphy (Toby et al., 2019). Key of this approach was to derive an autogenic threshold function (ATF) that separates conditions of signal transfer from signal shredding. In this study the same authors develop their approach further in order to apply it to ancient sediment routing systems (SRS). First, they theoretically derive how the ATF can be determined based on a limited number of parameters, which can be quantified or estimated from field measurements of ancient SRSs. Second, they validate their ideas by testing them against laboratory experiments performed under controlled boundary conditions. Last, they apply the methods to two exemplary field cases in Greece and the Pyrenees. This new method, for the first time, allows to quantify which magnitude-frequency combinations of sediment flux signals can be preserved and are, thus, usable for reconstruction of past environmental conditions.

The presented approach of determining an ATF of ancient SRS as "a tool to predict the presence (or absence) of any magnitude-period combination of input sediment flux signal in strata" is novel and a substantial contribution to the community, as we currently lack a method to evaluate which parts of the stratigraphic record can reliably be used for signal inversion (i.e. reconstruction of past environmental conditions). Also, the outcome that a sediment flux signal might not be preserved in the stratigraphy of an upper segment of a SRS, but can still be detected in stratigraphy further downstream is an important finding for any field study using stratigraphic archives. As such, the presented approach is timely and highly significant to the community.

In the presented study, the authors propose a pathway how their previously proposed method (Toby et al., 2019) can be applied to ancient sediment routing systems. They validate their approach using previously published experiments, before applying it to two field sites by using previously published measurements from those sites. This approach is scientifically sound. The presented study does not present any new datasets, but a new approach to evaluate the reliability of stratigraphic data for signal inversion. The current version does not contain any supplement. I have only two main comments, one about a better documentation of the validation and one on a more detailed explanation of the derivation of their proposed method, which is currently presented in a fairly minimalist way. Both of these points can be addressed by including a supplement file. All other comments mainly regard the improvement of clarity.

The presented approach was validated with experimental laboratory data. But the validation would benefit from a more detailed documentation. In Table 1 and lines 102-103 and 107-108 the authors compare values for the two parameters, b and $Q0^*$, derived from their proposed method and from laboratory experiments. But how were those experimental values derived? How was A (area) derived? What kind of experiments have been used (as it were different experiments as those presented in figure 1)? How does the data from those validation experiments look like? I think including a more detailed documentation of those validation experiments including the distribution of the data and how the values of b and $Q0^*$ were derived from those experiments would help the reader to follow the line of argument (especially the differences of b values derived from theory and experiments) and also to eventually apply this method to other datasets. Such a detailed documentation could maybe be added as a supplement file.

The greatest weakness of the approach is currently the discrepancy between theoretical and experimental b values (lines 102-103). The authors argue that natural systems, like the Mississippi River, behave differently than experimental systems (lines 104-106). A second field example would

help to strengthen this statement, and help to convince that the approach works despite the differences in b values in the validation.

Overall, the authors put much emphasis on discussing their proposed approach including any potential limitations, which is highly appreciated. However, the theoretical derivation of the method (lines 64-97) is quite minimalist. It includes several abbreviations, many equations within the text and requires detailed knowledge on a range of concepts, e.g. the AFT (Toby et al., 2019) and the compensation timescale, T_c . I think a little more explanation of the derivation would make this work accessible by a larger community. A more detailed derivation of the concepts (including all the equations) with explanations of which parameters need to be measured in the field could be included in the supplement. For example, I was confused as in lines 78-81 it is stated that "Two or more combinations of Q_a^* and T^* are sufficient to constrain the value of Q_0^* and b ..." and I was expecting that when applying the proposed method to field data, the aim was to find at least two combinations of Q_a^* and T^* and perform a regression analysis in order to determine the AFT. Instead, the presented approach described in the following paragraphs aims at directly estimate b and Q_0^* . As such, a more detailed explanation of the presented approach (in the supplement) would make it easier to follow the proposed approach.

Figure 1 is references in the introduction (lines 49-50) as an explanation that the AFT can be described by an exponential decay function (lines 49-50). However, especially panel a in figure 1 needs further explanation. Although I have worked with experimental data before, I find it hard to understand what can actually be seen here. Maybe a few labels of the experiment would already help. The separation of ESS is not well explained. Where does it come from?

The evaluation of whether a certain magnitude-period combination of input sediment flux signal is preserved in stratigraphy comes with two groups of uncertainties: (1) uncertainties related to the AFT and (2) uncertainties about past signal period and magnitude and absolute values. While error bars are included for the signals themselves (figure 3b and 6), it might be useful to also include error bars or uncertainties to the AFT. Right now, the authors address this problem by plotting several AFT curves (lines 124-125, figure 3b), and it is kind of tempting to argue that a certain signal is safely preserved in stratigraphy as long as the signals falls above the AFT. However, eventually we are interested in probabilities of how likely it is that a certain stratigraphic section as formed due to allogenic or autogenic processes. Including uncertainties to the AFT (based on uncertainties in b and Q_0^*) could be a first step in this direction.

Another suggestion (but not expected to be included here) would be the development of some kind of GUI. Ideally, any kind of environmental reconstruction from stratigraphy is preceded by such a quality assessment as presented in this study. A standardized, easy way to perform this would be of great benefit to the community.

Line-by-line comments

- Line 78 7: Doesn't it need to be a minimum of three? For two datapoints a straight line would be sufficient.
- Line 87: Constant through space?
- Line 102: Reduces TO 0.05 Q_a ?
- Lines 101-103: Here, it would be nice to see the experimental data, i.e. a time series of Q_a^* including the exponential regression and associated errors (see comment above). How where the needed parameters derived from this time series?
- Lines 148-149: Does this mean that if there is no bypass (= final sink of any SRS) signals will always be preserved here? Or in other words as long as any sediments makes it to the final sink, it will plot above the ATF in the signal preservation domain?
- Line 168: 'There are' or 'The' instead of 'There'.
- Lines 172-174: But isn't this a matter of system size, as T_{eq} increases with system size and T_c does not?

- Lines 233-236: This conclusion I cannot follow, why would this be the case?
- Line 245: Small e in 'Environmental'?
- Line 340: Remove open bracket before KDS.

Reviewer #2:

Remarks to the Author:

This paper provides some important new dynamic stratigraphic insights, and is of considerable significance for a range of Geoscientists. The authors offer a new concept and field method for estimating autogenic response thresholds and for differentiating between environmental signals and autogenic signals in the interpretation of alluvial stratigraphy. The authors claim in the introduction that 'their method will help geoscientists explore stratigraphy for Earth's response signatures of environmental forcing', but it is a weakness in the paper that they deal almost only with fluvial strata, essentially stopping any detailed analysis before the sediment reaches the shoreline, shelf or deep-water areas. This is re-enforced by their choice of field examples, both in a setting that is very proximal, in alluvial fan and fluvial landscapes. This however does not detract from their achievement and analysis in the fluvial systems; they should simply make this clear at an early stage of the paper, instead of allowing the reader to think that the new method can be applied to a large variety of sedimentary settings. As far as I can see, the theoretical development, work flow, experimental validation and application of the autogenic threshold function to ancient field systems is well argued. I have made some additional comments in the text, the most important of which are:

1. The two papers (2008, 2010) you cite in line 33 are not really the fundamental papers dealing with autogenic responses, how about the perspective JSR paper on auto-stratigraphy by Muto et al 2007 or the earlier papers by Muto
2. Be more clear about the term 'signals'. In the introduction, a short list of the types of signals that commonly become shredded during sediment transport would be useful to readers. Also make it clear that that it is mainly sediment flux signals that your analysis focusses on.
3. There is brief mention (lines 35-40) of the basin-response time scale approach to the problem of signal propagation. Why not also mention another earlier approach, albeit 20 years ago, that made use of the auto-retreat function to get a more correct view (at that time) of sediment flux and accommodation (Muto & Steel, 2002, BR)
4. The 6 or so pages of the Discussion are tightly written and would benefit from the addition of 2-3 subheading to make it more reader friendly.

I recommend publication of this paper if the above moderate revision can be done

Response to reviewer comments on *Morphodynamic limits to environmental signal propagation across landscapes and into strata* by Toby et al.

Reviewer #1 (Remarks to the Author):

Dear Editors and Authors,

Toby and colleagues have recently proposed a novel approach to quantify whether or not certain magnitude-frequency combinations of sediment flux signals are likely to be preserved in stratigraphy (Toby et al., 2019). Key of this approach was to derive an autogenic threshold function (ATF) that separates conditions of signal transfer from signal shredding. In this study the same authors develop their approach further in order to apply it to ancient sediment routing systems (SRS). First, they theoretically derive how the ATF can be determined based on a limited number of parameters, which can be quantified or estimated from field measurements of ancient SRSs. Second, they validate their ideas by testing them against laboratory experiments performed under controlled boundary conditions. Last, they apply the methods to two exemplary field cases in Greece and the Pyrenees. This new method, for the first time, allows to quantify which magnitude-frequency combinations of sediment flux signals can be preserved and are, thus, usable for reconstruction of past environmental conditions.

The presented approach of determining an ATF of ancient SRS as “a tool to predict the presence (or absence) of any magnitude-period combination of input sediment flux signal in strata” is novel and a substantial contribution to the community, as we currently lack a method to evaluate which parts of the stratigraphic record can reliably be used for signal inversion (i.e. reconstruction of past environmental conditions). Also, the outcome that a sediment flux signal might not be preserved in the stratigraphy of an upper segment of a SRS, but can still be detected in stratigraphy further downstream is an important finding for any field study using stratigraphic archives. As such, the presented approach is timely and highly significant to the community.

Yes. We agree.

In the presented study, the authors propose a pathway how their previously proposed method (Toby et al., 2019) can be applied to ancient sediment routing systems. They validate their approach using previously published experiments, before applying it to two field sites by using previously published measurements from those sites. This approach is scientifically sound. The presented study does not present any new datasets, but a new approach to evaluate the reliability of stratigraphic data for signal inversion. The current version does not contain any supplement.

Thanks, we now include a new Supplementary Information document.

I have only two main comments, one about a (1) better documentation of the validation and one on (2) a more detailed explanation of the derivation of their proposed method, which is currently presented in a fairly minimalist way. Both of these points can be addressed by including a supplement file. All other comments mainly regard the improvement of clarity.

We now include this information in a new Supplementary Information document.

The presented approach was validated with experimental laboratory data. But the validation would benefit from a more detailed documentation. In Table 1 and lines 102-103 and 107-108 the authors compare values for the two parameters, b and Q_0^* , derived from their proposed method and from laboratory experiments. But how were those experimental values derived? How was A (area) derived? What kind of experiments have been used (as it were different experiments as those presented in figure 1)? How does the data from those validation experiments look like? I think including a more detailed documentation of those validation experiments including the distribution of the data and how the values of b and Q_0^* were derived from those experiments would help the reader to follow the line of argument (especially the differences of b values derived from theory and experiments) and also to eventually apply this method to other datasets. Such a detailed documentation could maybe be added as a supplement file.

We now have a 'supplementary material' document that contains this information.

The greatest weakness of the approach is currently the discrepancy between theoretical and experimental b values (lines 102-103). The authors argue that natural systems, like the Mississippi River, behave differently than experimental systems (lines 104-106). A second field example would help to strengthen this statement, and help to convince that the approach works despite the differences in b values in the validation.

We present information on several river systems in Table S2 Supplementary Material. We demonstrate that the avulsion timescales (\$T_A\$ ) for several river systems are as follows: Mississippi River, \$T_A \approx 0.007T_c^{8,27}\$; Orinoco River, \$T_A \approx 0.027T_c\$; Paranà River, \$T_A \approx 0.006T_c\$; Po River, \$T_A \approx 0.06T_c\$; Rhine delta, \$T_A \approx 0.03T_c\$; Yellow River, \$T_A \approx 0.018T_c\$; and Yellow River, delta \$T_A \approx 0.002T_c\$.

Overall, the authors put much emphasis on discussing their proposed approach including any potential limitations, which is highly appreciated. However, the theoretical derivation of the method (lines 64-97) is quite minimalist. It includes several abbreviations, many equations within the text and requires detailed knowledge on a range of concepts, e.g. the AFT (Toby *et al.*, 2019) and the compensation timescale, T_c . I think a little more explanation of the derivation would make this work accessible by a larger community. A more detailed derivation of the concepts (including all the equations) with explanations of which parameters need to be measured in the field could be included in the supplement.

Further explanation of the theoretical derivation of the method and the application of the method, including further explanation of the AFT and \$T_c\$, is presented in the new Supplementary Material section.

For example, I was confused as in lines 78-81 it is stated that “Two or more combinations of Q_0^* and T^* are sufficient to constrain the value of Q_0^* and b ...” and I was expecting that when applying the proposed method to field data, the aim was to find at least two combinations of Q_0^* and T^* and perform a regression analysis in order to determine the AFT. Instead, the presented approach described in the following paragraphs aims at directly estimate b and Q_0^* . As such, a more detailed explanation of the presented approach (in the supplement) would make it easier to follow the proposed approach.

Thank you. We now include a supplementary material that contains this information.

Figure 1 is references in the introduction (lines 49-50) as an explanation that the AFT can be described by an exponential decay function (lines 49-50). However, especially panel a in figure 1 needs further explanation. Although I have worked with experimental data before, I find it hard to understand what can actually be seen here. Maybe a few labels of the experiment would already help. The separation of ESS is not well explained. Where does it come from?

Thank you. We have added annotations and explanations to figure 1a, which we feel has improved it considerably.

To explain further the separation of ESS we add: “The ATF framework predicts that apart from signals that exceed the ATF, high-frequency ($T^* \ll 1$) sediment flux signals with a particularly high rate of flux change (region ESS; Fig. 1b) can induce an active layer or geomorphic response⁷. This happens because autogenic surface processes cannot redistribute sediment across the active layer at the required rate to maintain an equilibrium landscape with respect to the signal of sediment flux. Therefore this class of signal, especially if $Q_a^* > 2.5$ (Fig. 1b), are still transmitted through the active layer of a sediment routing system.”

The evaluation of whether a certain magnitude-period combination of input sediment flux signal is preserved in stratigraphy comes with two groups of uncertainties: (1) uncertainties related to the AFT and (2) uncertainties about past signal period and magnitude and absolute values. While error bars are included for the signals themselves (figure 3b and 6), it might be useful to also include error bars or uncertainties to the AFT. Right now, the authors address this problem by plotting several AFT curves (lines 124-125, figure 3b), and it is kind of tempting to argue that a certain signal is safely preserved in stratigraphy as long as the signals falls above the AFT. However, eventually we are interested in probabilities of how likely it is that a certain stratigraphic section as formed due to allogenic or autogenic processes. Including uncertainties to the AFT (based on uncertainties in b and Q_0^*) could be a first step in this direction.

Thank you for this comment. We now report errors on field parameter values in Table 2 and use these to propagate errors to: 1) the ATF curves for field sites; and 2) data points on each ATF diagram in figure 6. For figure 3b we construct ATFs for a conservative range of Q_{in} values, i.e. $Q_{acc}/Q_{in} = 0.5 \pm 0.4$, which we believe captures any uncertainty in calculation of Q_{in} and demonstrates that these short, syn-rift, coarse-grained systems are a ideal site to store higher frequency climate signals.

Another suggestion (but not expected to be included here) would be the development of some kind of GUI. Ideally, any kind of environmental reconstruction from stratigraphy is preceded by such a quality assessment as presented in this study. A standardized, easy way to perform this would be of great benefit to the community. Thank you for the suggestion.

Line 78 7: Doesn't it need to be a minimum of three? For two datapoints a straight line would be sufficient. Two data points can also be used to construct an exponential line. Recalling that in semi-log space an exponential appears as a straight line, and thus if one wished, they could define this fit

in a linear domain defined by T^* and $\log(Q_a^*)$ and then make an easy transformation of the fit parameters to the exponential function. However, this is just for demonstration; the point is that in practice this function can be defined by a minimum of two known points. Given input values for Q_a^* at $T^* = 0$ (i.e. Q_0^*) and at $T^* = 1$ (for $T_{95}/T_c = 1$), exponent b takes a constant value of $b \approx 3$ (or 3.15) in the dimensionless equation: $b = -\ln(0.05)/T_{95}^*$, where $T_{95}^* = T_{95}/T_c$.

Line 87: Constant through space? We now write “constant through time”

Line 102: Reduces TO 0.05Qa? Changed to “reduces to 0.05Q₀*”. Thanks.

Lines 101-103: Here, it would be nice to see the experimental data, i.e. a time series of Qa* including the exponential regression and associated errors (see comment above). How where the needed parameters derived from this time series?

The plots of time series of Qa* including the exponential regression is now presented in the supplementary information file. We also provide a fuller account of how key parameters were derived. Thank you.

Lines 148-149: Does this mean that if there is no bypass (= final sink of any SRS) signals will always be preserved here? Or in other words as long as any sediments makes it to the final sink, it will plot above the ATF in the signal preservation domain?

In practice this is the case, yes. We note that our experimental ATF is a theoretical limit for signal preservation. As such, it divides the domain into two regions: a region where a signal should be theoretically shredded; a region where a signal should theoretically be persevered. However, in order to identify the signal you may require an impeccable (as yes unachievable) and high resolution temporal constraints. Thus, in the ultimate sink, if one could track the age of the sediment volumes deposited with infinite precision, theoretically you could define a time series of changes of sediment flux to the sink as no sediment bypasses this site (i.e. can undergo bypass and release out of the sink). In practice, limits on our geochronometers would be the hurdle to overcome in these sinks.

Line 168: ‘There are’ or ‘The’ instead of ‘There’. Change made.

Lines 172-174: But isn’t this a matter of system size, as T_{eq} increases with system size and T_c does not? This is an interesting comment. River length or system size L is dependent on sediment flux, Q_s , and subsidence, r (Whipple and Trayler, 1996; Duller *et al.*, 2010). Channel depth, H , is also implicitly dependant on L through water discharge (Leopold *et al.*, 1964) but the precise relationship will depend on whether the river system is tributive or distributive. The value of T_c explicitly depends on subsidence, r , and H ($T_c = r/H$), which means T_c will also change as L increases (as H changes). So we cannot assume T_c remains constant while system size changes. We contend that T_{eq} and T_c are affected by the same fundamental processes that contribute to the construction of an equilibrium landscape. Future work will help understand the relationship between T_c and T_{eq} .

Lines 233-236: This conclusion I cannot follow, why would this be the case? We add the following text to the manuscript to clarify: “The ATF framework predicts that apart from signals that exceed the ATF, high-frequency ($T^* \ll 1$) sediment flux signals with a particularly high rate of flux change (region ESS; Fig. 1b) can induce an active layer or geomorphic response. This happens because autogenic surface processes, that act to redistribute sediment across the active layer, cannot

respond quickly enough to attain equilibrium with the sediment flux signal. As a consequence this class of signal, especially those with $\approx Qa^* > 2.5$ (Fig. 1b), are still efficiently transmitted through the active layer of a SRS. Furthermore we add the following text to section T2 in the supplementary materials: “Toby et al (2019) found that below the ATF in the signal shredding regime, some sediment supply signals induce a geomorphic response but not a stratigraphic response; and some produce neither a geomorphic or stratigraphic response. Toby et al. (2019) show that the ability of sediment supply signals to produce a geomorphic surface response decreased as the period of sediment supply increased and defined a signal acceleration threshold to delineate these two different responses (diagonal dashed line, figure 1b). A rapid change in sediment supply, even if only small in total magnitude (such as a large flood), can trigger a transient geomorphic response at the Earth's surface, as the system cannot respond quickly enough to attain equilibrium with the sediment supply forcing. However it is not transferred to strata as the period of the sediment supply signal is short, and cannot produce a thick enough sedimentary pile to escape subsequent reworking by autogenic processes within the active surface layer. The region of the ATF in figure 1a defined as ESS (Earth surface signal) with diagonal fill represents the space where a geomorphic response or active layer response of this kind is likely. If the active layer reacts to a sediment supply signal, or periodic sediment supply cycles, then they are by definition more likely to convey the signal through the landscape or to the next segment in the sediment routing system, even though these signals are unlikely to be preserved in strata. It may indeed be the case that signals of this kind (i.e. those that occupy ESS space on fig. 1b) attenuate rapidly down-system however, depending on initial signal magnitude; it is possible that a signal could plot above ATF of the next segment and therefore into strata.”

Line 245: Small e in ‘Environmental’? Change made.

Line 340: Remove open bracket before KDS. Change made.

Reviewer #2 (Remarks to the Author):

This paper provides some important new dynamic stratigraphic insights, and is of considerable significance for a range of Geoscientists. The authors offer a new concept and field method for estimating autogenic response thresholds and for differentiating between environmental signals and autogenic signals in the interpretation of alluvial stratigraphy.

The authors claim in the introduction that ‘their method will help geoscientists explore stratigraphy for Earth’s response signatures of environmental forcing’, but it is a weakness in the paper that they deal almost only with fluvial strata, essentially stopping any detailed analysis before the sediment reaches the shoreline, shelf or deep-water areas. This is re-enforced by their choice of field examples, both in a setting that is very proximal, in alluvial fan and fluvial landscapes. This however does not detract from their achievement and analysis in the fluvial systems; they should simply make this clear at an early stage of the paper, instead of allowing the reader to think that the new method can be applied to a large variety of sedimentary settings.

Our emphasis is placed on fluvial channelized systems and associated strata as this was the experimental environment used to define and test the ATF theory in the initial work of Toby et al. (2019). However we anticipate that any depositional system with a channel (e.g. shallow marine, deepwater etc) can be treated in much the same way; not just fluvial. Furthermore, the definition of the compensation timescale as we define it necessarily utilizes channel depth (H_c) as the largest roughness length scale. However the compensation timescale is applicable to any environment, not just channelized environments, where a topographic roughness features of length scale l_{max} (e.g. turbulent sweeps, ripples, dunes) migrate over a sediment surface that is net depositional over the long term. Therefore we anticipate that an ATF can be constructed for a range of environments and at a variety of scales, not just for channelized environments at the basin architecture scale.

To clarify this we add the following to the main manuscript:

“ T_c can also be thought of as the maximum time necessary to bury a particle to a depth that is no longer susceptible to erosion from a maximum, vertical topographic roughness length scale, l_{max} (i.e. $T_c = l_{max}/r$) at the Earth’ surface (Wang et al., 2011; Straub & Esposito, 2013; Straub & Foreman, 2018). Therefore the compensation timescale is applicable to any environment, not just channelized environments, where topographic roughness features migrate over a sediment surface that is net depositional over the long term. For example, even in relatively quiescent lake and abyssal plain environments, the migration of roughness features like ripples, dunes, and bars will result in autogenic fluxes that will saturate at a commensurate timescale, τ_c . However, we note the detrimental effects of autogenic processes for the preservation of environmental signals in strata will be most pronounced in channelized (e.g. fluvial and deepwater channel-fan) environments.”

“While the autogenic threshold function (ATF) was constructed with channelized environments in mind, we note that all depositional environments contain an element of autogenic sediment flux, which gives rise to temporally incomplete strata at the finest timescales of discretization. Therefore we anticipate that an ATF can be constructed for other environments at different temporal and physical scales¹⁷”

As far as I can see, the theoretical development, work flow, experimental validation and application of the autogenic threshold function to ancient field systems is well argued. I have made some additional comments in the text, the most important of which are:

1. The two papers (2008, 2010) you cite in line 33 are not really the fundamental papers dealing with autogenic responses, how about the perspective JSR paper on auto-stratigraphy by Muto et al 2007 or the earlier papers by Muto.

Thank you. We now add ‘Muto, T., Steel, R. & Swenson, J. Autostratigraphy: A Framework Norm for Genetic Stratigraphy. *Journal of Sedimentary Research* 77, 2-12 (2007)’ to correct this oversight.

2. Be more clear about the term ‘signals’. In the introduction, a short list of the types of signals that commonly become shredded during sediment transport would be useful to readers.

This has been addressed at the outset in the introduction: “An environmental signal is a physio-bio-chemical attribute of the Earth’s surface and/or sediments that can be linked to a particular environmental forcing (tectonic, climatic, sea level, anthropogenic)1. Environmental signals can

represent any time series of data (e.g. C and O isotope data, palynological data, bulk compositional data, grain size data, ichnological data etc)".

Also make it clear that that it is mainly sediment flux signals that your analysis focusses on. To the introduction we have now added: "A key measurable attribute that links environmental forcing, Earth surface processes and the sedimentary record is sediment flux, and we focus our efforts on this attribute."

3. There is brief mention (lines 35-40) of the basin-response time scale approach to the problem of signal propagation. Why not also mention another earlier approach, albeit 20 years ago, that made use of the auto-retreat function to get a more correct view (at that time) of sediment flux and accommodation (Muto & Steel, 2002, BR)

Thank you. We use the reference of Paola *et al.* (1992) as this work derives the diffusion equation from first principles and really serves as a clear point of reference for an interested reader wising to delve into its formulation. We see no added benefit to including the auto-retreat function (e.g. Muto and Steel, 2002), which may confuse the reader.

4. The 6 or so pages of the Discussion are tightly written and would benefit from the addition of 2-3 subheading to make it more reader friendly.

Thank you. We have added several headings to help direct the reader:

- Key uncertainties in the field-approximated autogenic threshold function (ATF)
- Temporal limits of signal propagation through sediment routing systems
- The critical role of the Earth surface active layer
- Sediment flux signals in deep marine strata
- Sediment routing system sensitivity.

I recommend publication of this paper if the above moderate revision can be done.

All revisions have now been done.

Reviewers' Comments:

Reviewer #1:

Remarks to the Author:

I have re-read the manuscript 'Morphodynamic limits to environmental signal propagation 1 across landscapes and into strata' by Toby et al. The authors have carefully addressed all issues raised by both reviewers and adjusted the manuscript accordingly. In particular, I appreciate the effort the authors have put in preparing the supplementary file including a useful summary of concepts and more details on the experiments and ATF estimations. I only have a few very minor remaining points and suggest publication after these have been solved.

The authors have added a supplementary file containing six subsections (although the last two are both called T5) providing additional information about the framework, experiments and how to estimate the ATF. But the supplementary file is nowhere cited in the text. I suggest (if compatible with the journal style) to refer to the six subsections individually in the manuscript to provide the reader additional information when most appropriate.

Lines 26-28: I do not fully agree with the definition of an 'environmental signal', which has been added here upon request of reviewer 2. The authors define an environmental signal as "a physio-bio-chemical attribute of the Earth's surface and/or sediments that can be linked to a particular environmental forcing (tectonic, climatic, sea level, anthropogenic)." I would argue that a signal is not an attribute, but instead a change in the attribute (or parameter that can be measured at the surface or in sediments) through time. If the attribute value is constant through time, I would not call this a signal. I also refer the authors to a recently published review in which we have addresses exactly this issue in more detail (Tofelde et al., 2021, *Frontiers in Earth Science*).

Line 278: Either 'this class of signal' or 'these classes of signals'?

Line 291: Is 'dissipated transformed' grammatically correct?

I apologize again for the delay in returning the manuscript.

Reviewer #2:

None

REVIEWERS' COMMENTS

The authors have added a supplementary file containing six subsections (although the last two are both called T5) providing additional information about the framework, experiments and how to estimate the ATF. But the supplementary file is nowhere cited in the text. I suggest (if compatible with the journal style) to refer to the six subsections individually in the manuscript to provide the reader additional information when most appropriate.

1. We have corrected the subsection headings in the error in the supplementary information. There were two 'T5' sections. The second 'T5 section' is now labelled as 'T6'.
2. Thanks. There is now reference to all subsections of the supplementary information in the main document.

Lines 26-28: I do not fully agree with the definition of an 'environmental signal', which has been added here upon request of reviewer 2. The authors define an environmental signal as "a physio-bio-chemical attribute of the Earth's surface and/or sediments that can be linked to a particular environmental forcing (tectonic, climatic, sea level, anthropogenic)." I would argue that a signal is not an attribute, but instead a change in the attribute (or parameter that can be measured at the surface or in sediments) through time. If the attribute value is constant through time, I would not call this a signal. I also refer the authors to a recently published review in which we have addresses exactly this issue in more detail (Tofelde et al., 2021, Frontiers in Earth Science).

Thank you. Yes, you are correct. This was not our intention and now we reword to account for the change in the attribute, as you say. We now write: "An environmental signal is a change in any physio-bio-chemical attribute of the Earth's surface and/or sediments in response to environmental forcing (tectonic, climatic, sea level, anthropogenic)¹. Environmental signals may be found in time series of, for example, C and O isotope data, palynological, ecological, petrophysical or petrographic data, or indeed any measurable, observable time-dependent property linked to environmental forcing"

Line 278: Either 'this class of signal' or 'these classes of signals'?

Corrected

Line 291: Is 'dissipated transformed' grammatically correct?

Corrected

I apologize again for the delay in returning the manuscript.

No problem. Thanks for taking the time on behalf of Nature publishing and the authors.